# Symmetry-Protected Geometric Separation for Verified Algebraic Reasoning

## Abstract

Many reasoning problems are defined by algebraic invariance: multiple surface forms represent the same computation, and correctness must remain stable under semantics-preserving rewrites and increasing compositional depth—a regime where standard sequence models can exhibit severe drift. We study a mechanism that targets this setting through explicit operator composition. Holonomic Networks associate tokens with learned geometric actions in $SO(d)$ and implement computation by their ordered composition (holonomy). We formalize the Holonomic Network variable-binding swap-program instantiation in this operator language and show that the same compositional engine supports multiple decoders. Replacing variable readout by an identity-energy decoder yields a unified verifier for algebraic word identity (whether a sequence of generators evaluates to the identity element), scoring a word by the normalized Frobenius distance of its composed operator to the identity. We instantiate the template for finite Coxeter words in $S_{32}$ and infinite braid words in $B_8$, organizing specialization through a compact set of enforcement modes: hypothesis-class restriction to $SO(d)$, architectural tying where available (inverse-by-transpose), symmetry-alignment losses with anti-collapse guardrails, and oracle-audited evaluation. Trained only on short words ($L \leq 50$), the resulting verifiers achieve perfect sampled classification under a $100\times$ length extrapolation to $L = 5000$ in both settings. In the Coxeter case, identity and non-identity energies remain separated by a persistent positive gap, yielding TPR = 1 and FPR = 0 on the reported evaluations. For braid words, an independent SageMath audit confirms agreement with the oracle on a 14,000-example evaluation dump with zero detected label mismatches.

## 1 Introduction

Neural networks can model sequences at scale, yet exact compositional behavior over long horizons remains difficult to obtain reliably (Lake & Baroni, 2018; Hupkes et al., 2020; Veličković & Blundell, 2021; Veličković et al., 2022). In many domains, correctness is not tied to local token statistics but to the semantics of a long composition of operations. A sharp operational criterion is rewrite invariance: multiple surface forms represent the same computation, and a correct system must return identical outputs across broad families of semantics-preserving rewrites while remaining stable as compositional depth increases. The algebraic word identity task, known in group theory as the word problem, provides a clean testbed for this criterion. It asks whether a word over generators evaluates to the neutral identity element, a property governed by global algebraic cancellations rather than local token frequencies. It provides a stringent stress test for whether continuous representations can sustain exact multi-step composition at depths far beyond training without catastrophic drift. We focus on non-abelian settings, where order is semantic ($AB \neq BA$), so correctness depends on the full sequence of composed actions rather than token counts. The label is invariant under relation-preserving rewrites, depends on global cancellations and noncommutativity, and admits trusted oracle evaluation. In addition, an identity-energy decoder makes the computation auditable: generalization is reflected not only in thresholded accuracy but in the persistence of an explicit geometric margin between identity and non-identity words under extrapolation.

This perspective connects directly to term rewriting and equational reasoning, where semantics is specified by rewrite systems and equivalence is defined by closure under relation-preserving transformations (Baader & Nipkow, 1998; Terese, 2003; Knuth & Bendix, 1970). It also aligns with geometric deep learning, which treats invariance/equivariance as a design principle and views compatible local actions as the primitive from which representations are constructed (Cohen & Welling, 2016; Bronstein et al., 2021; Kondor & Trivedi, 2018). In sequence form, a token-indexed family of orthogonal actions can be interpreted as defining a discrete connection over positions, with the composed operator as the associated holonomy (Bronstein et al., 2021; 2017). This interpretation places the present work in the same symmetry-first ecosystem as gauge-equivariant architectures (Cohen et al., 2019) and flow-equivariant recurrent models based on Lie actions (Keller, 2025).

Holonomic Networks propose a concrete instantiation of this operator-composition view (Sung, 2026). Each token selects a learned geometric action, computation is carried by their ordered composition, and a downstream decoder reads out the desired quantity. The Holonomic Network variable-binding setting realizes this as latent-state transport followed by a standard classifier; no symbolic executor is required. This operator-learning perspective also clarifies the contrast with additive recurrent updates $h_t = f_\theta(h_{t-1}, x_t)$: when the effective operation is entangled with the state and nonlinearity, small approximation errors can accumulate with depth. Holonomic Networks make the composed operation explicit in the hypothesis class and stabilize long horizons through near-isometric actions in $SO(d)$ (and, in variable binding, per-step renormalization), relating to work on orthogonal/unitary recurrent dynamics (Arjovsky et al., 2016; Wisdom et al., 2016). Complementary approaches to long-horizon sequence modeling, including attention-based and state-space formulations, pursue stability through different inductive biases (Vaswani et al., 2017; Gu et al., 2022); our focus is the holonomic operator-composition route.

This paper makes three contributions. First, we formalize the Holonomic Network mechanism in operator language on its variable-binding task and isolate its decoder modularity. Second, we show that replacing variable readout by an identity-energy decoder yields a unified template for algebraic word identity verification. Third, we instantiate the template across a deliberate escalation of algebraic difficulty—from the finite Coxeter presentation of $S_{32}$ to the infinite braid group $B_8$—and evaluate length extrapolation under hygiene-aware protocols, including exact oracle labeling (Coxeter) and an independent SageMath audit (The Sage Developers, 2026) on exported prediction dumps (braids). The resulting systems exhibit symmetry-protected geometric separation in an operational sense: identity and non-identity examples occupy separated energy regimes with a persistent gap under substantial length extrapolation.

The remainder of the paper proceeds as follows. Section 2 presents the Holonomic Network principle, the decoder swap, and the operational meaning of symmetry-protected geometric separation. Section 3 describes the unified neural template for algebraic word identity. Section 4 details the Coxeter and braid instantiations. Section 5 reports results and robustness analyses, and Section 6 discusses implications and extensions.

## 2   The Holonomic Network Principle

This section formalizes the conceptual bridge that organizes the paper. We first restate the original Holonomic Network construction for variable binding in a direct mathematical form, emphasizing how compositional semantics is realized through learned operator composition and standard neural operations. We then show that the transition from variable binding to algebraic word identity is a decoder-level change rather than a change in the underlying compositional computation. Figure 1 summarizes the shared composition template and the decoder swap studied in this paper. Finally, we define the operational meaning of symmetry-protected geometric separation used throughout the paper.

### 2.1   Holonomic Network for variable binding via operator composition

The Holonomic Network variable-binding task takes as input a swap program: a sequence of elementary swaps over a finite set of values (or memory slots), and the target is the value occupying a distinguished slot after executing the swap sequence (Sung, 2026). We summarize this setting only to isolate the compositional operator mechanism that underlies length generalization and that is reused later with a different decoder. Let $V = \{0, \dots, N-1\}$ denote the value set and let $\mathcal{A}$ denote a token vocabulary of size $K$ indexing elementary

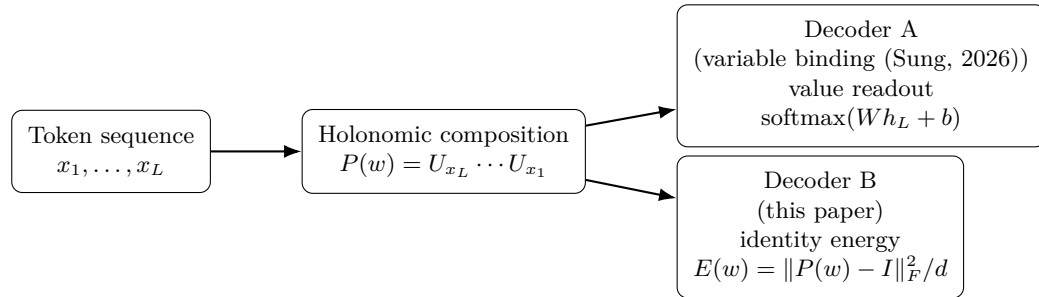

Figure 1: **Holonomic Network principle as a reusable inference template.** The holonomic composition core is shared across tasks; the decoder can be swapped (variable readout vs. identity energy), and algebraic structure is specialized through a small set of enforcement modes (geometry, tying, alignment, auditing).

swaps. A swap program is a sequence $w = (x_1, \ldots, x_L)$ with $x_t \in \mathcal{A}$, and the supervision target $y_{\text{vb}}(w) \in V$ is obtained by symbolic execution of the sampled swap program. The model does not execute swaps; it learns to represent the compositional semantics of the program in a continuous geometry.

Holonomic Networks associate each token $a \in \mathcal{A}$ with a learned orthogonal action $U_a \in SO(d)$, parameterized by exponentiating a skew-symmetric matrix:

$$A_a = M_a - M_a^\top, \qquad U_a = \exp(A_a) \in SO(d), \tag{1}$$

where $M_a \in \mathbb{R}^{d \times d}$ is unconstrained. Given value embeddings $\{e_v\}_{v \in V} \subset \mathbb{R}^d$, the computation transports a latent probe state by sequential composition,

$$h_t = \mathcal{N}(U_{x_t} h_{t-1}), \qquad t = 1, \ldots, L, \tag{2}$$

with $\mathcal{N}(z) = z/\|z\|_2$, and decodes the output using a standard linear classifier head,

$$p_\theta(y_{\text{vb}} \mid w) = \text{softmax}(W h_L + b), \tag{3}$$

trained end-to-end with cross-entropy. The forward computation uses only learned linear actions, normalization, and a conventional classifier readout; symbolic execution is confined to data generation for supervision.

In this setting, supervision is multi-class and acts directly on the transported state. The resulting gradient signal is dense enough to shape the operator geometry without introducing explicit algebraic relation penalties in the training objective.

This design makes the compositional object explicit. In this view, the holonomy—the composed operator determined by the token sequence—is the semantic object associated with the program.

Viewed through the physical lens developed in (Sung, 2026), the token-indexed operators define a discrete gauge connection and the forward pass performs parallel transport, so the ordered product is a holonomy observable. The per-step renormalization is then a gauge-fixing or conditioning choice that keeps the transported state on a well-behaved section (the unit sphere) without changing the semantic content carried by the composed action. Each token selects an operator, and the cumulative effect of a sequence is governed

by multiplicative composition. As noted in the Introduction, this contrasts with additive recurrent updates $h_t = f_\theta(h_{t-1}, x_t)$, where approximation errors can accumulate with depth. Holonomic Networks instead represent the computation as composition of near-isometries in $SO(d)$. The per-step renormalization in (2) further stabilizes long compositions in finite precision while preserving directional information used by the readout.

Equation (2) includes renormalization after each step. In exact arithmetic, orthogonal actions preserve norm. During learning and in finite precision, small deviations can accumulate across long compositions; renormalization keeps the latent trajectory on the unit sphere and mitigates numerical drift without introducing any symbolic routine in the forward pass.

## 2.2 Decoder swap: from variable readout to identity energy

The mechanism is reusable because the composition principle is independent of the final decoder. The following components remain unchanged from the original Holonomic Network perspective:

1. tokens are represented by learned geometric transformations;

2. reasoning proceeds by ordered composition of those transformations;

3. the composed object is decoded by a simple downstream map.

In this sense, Holonomic Networks are best understood as a compositional neural template, not as a single task-specific architecture.

For algebraic word-identity verification, we replace the variable classifier decoder in (3) with an identity-energy decoder. Given a word $w = (x_1, \ldots, x_L)$, define the composed transformation

$$P(w) = U_{x_L} \cdots U_{x_1}, \tag{4}$$

and decode identity by the normalized Frobenius energy relative to the identity transformation:

$$E(w) = \frac{1}{d} \|P(w) - I\|_F^2 = 2 - \frac{2}{d} \operatorname{Tr}(P(w)). \tag{5}$$

Low energy indicates proximity to the neutral element; high energy indicates separation from it.

This decoder swap is conceptually small but mathematically important. In the Holonomic Network variable-binding task, the readout asks which value is represented by the transported latent state. In the present algebraic tasks, the readout asks whether the composed transformation itself is (approximately) the identity. The former is a discriminative readout over a transported vector; the latter is a geometric diagnostic on the composed operator. The composition mechanism, however, remains holonomic in exactly the same sense.

Variable binding and word identity share the same compositional skeleton:

$$\text{token sequence} \longrightarrow \text{composed transformation} \longrightarrow \text{task-specific readout.}$$

The Holonomic Network contribution is the middle arrow: a learned geometric composition mechanism that remains stable over long sequences. Once that mechanism is available, changing the readout from a classifier to an identity-energy criterion is a principled extension rather than a redesign.

Coxeter and braid instantiations require symmetry-aware regularization terms, carefully constructed training distributions, and (for braids) an external oracle-audit protocol. These components are essential for rigorous algebraic evaluation, but they do not alter the central inference computation in (4)–(5). The conceptual continuity with the original Holonomic Network framework is therefore exact at the level of neural inference, even when the supervision and validation pipelines differ substantially across tasks.

### 2.3 Operational meaning of symmetry-protected geometric separation

In this paper, the phrase symmetry-protected geometric separation is used in an operational empirical sense, with three components.

1. **Symmetry-aware alignment.** The learned generators are trained under regularization terms that encourage compatibility with algebraic relations (e.g., Coxeter or braid relations, commutation structure, and anti-collapse constraints). In contrast to the variable-binding setting, where a realizability condition yields a formal length-generalization guarantee via associativity and closure (Sung, 2026), these terms are soft and do not certify exact relation satisfaction. A natural next step is post-training certification that bounds relation residuals and the induced energy margin on the relevant input class.

2. **Geometric separation.** Identity and non-identity words are decoded by the same scalar energy $E(w)$ in (5). The phenomenon of interest is a persistent separation of the corresponding energy distributions, typically summarized by quantities such as

$$\max_{w \in \mathcal{D}_{\mathrm{id}}} E(w) \quad \text{and} \quad \min_{w \in \mathcal{D}_{\mathrm{non}}} E(w),$$

   and their difference (the energy gap) on a given evaluation set.

3. **Protection under extrapolation.** The term "protected" refers to the empirical stability of this gap under substantial extrapolation in word length relative to the training regime (in our experiments, training on short lengths and testing up to much larger lengths).

The variable-binding analysis in (Sung, 2026) provides theorem-level statements under a realizability condition; here the terminology summarizes an empirical separation phenomenon for identity-energy decoding.

This terminology captures the measured phenomenon without overstating what has been proved. The separation is geometric because it is defined by distances in the learned transformation space; it is symmetry-protected because it emerges reliably only when training aligns the geometry with algebraic relations; and it is protected in the empirical sense that the gap remains stable under large compositional depth. This is precisely the sense in which the subsequent Coxeter and braid results should be interpreted.

The Holonomic Network variable-binding setting does not use identity-energy decoding (5); it realizes compositional reasoning by holonomic transport in a learned continuous geometry with a standard neural readout. The present work preserves that principle and reveals an additional decoder regime in which the geometry itself becomes directly observable through an energy gap. This decoder-level modularity is the conceptual bridge between the original Holonomic Network formulation and the algebraic verifiers developed in the remainder of the paper. Unlike in variable binding, where a transported latent state may use per-step renormalization for numerical conditioning, identity-energy decoding depends only on products of (approximately) orthogonal operators in $SO(d)$, so no per-step renormalization is required in exact arithmetic.

## 3 Unified Neural Template for Algebraic Word Identity

Having isolated the Holonomic Network principle and the decoder swap from variable readout to identity energy, we now present the unified neural template used for algebraic word-identity verification. This section abstracts the shared inference and training structure that underlies both the Coxeter and braid instantiations. The purpose is to make explicit what is common across tasks—the holonomic neural computation—and what is deferred to task-specific design choices (relation sets, tokenization, data generation, and oracle protocols).

### 3.1 Token generators as learned orthogonal transformations

Let $\mathcal{A}$ denote a finite token vocabulary. Depending on the algebraic setting, tokens may represent generators only (as in the Coxeter case) or generators together with formal inverses (as in the braid case). We associate each token $a \in \mathcal{A}$ with a learned transformation $U_a \in SO(d)$.

As in the original Holonomic Network construction (Section 2), each $U_a \in SO(d)$ is parameterized by exponentiating a skew-symmetric matrix (Eq. (1)). This parameterization provides a smooth, trainable family of orthogonal transformations while preserving a strong geometric inductive bias, and keeps long-horizon composition within a single coherent operator geometry. In our instantiations, each token maps directly (via a learned linear table/embedding) to a skew-symmetric generator, so the forward computation contains no additional MLP head beyond the matrix exponential and operator products. For readers who jump directly to this template section, a self-contained restatement is recorded in Appendix C.

A shared geometric feature of Holonomic Network instantiations is that token actions live in $SO(d)$, so each learned operator has a canonical geometric inverse:

$$U_a^{-1} = U_a^\top. \tag{6}$$

In the braid instantiation, where inverse tokens appear explicitly, we enforce inverse consistency architecturally by parameter tying:

$$U_{a^{-1}} := U_a^\top. \tag{7}$$

In the Coxeter instantiation, inverse tokens are unnecessary because Coxeter generators are involutions; instead, we enforce the same structure relationally by an involution-alignment penalty that encourages $U_a^2 \approx I$, implying $U_a^{-1} \approx U_a$ (and hence $U_a^\top \approx U_a$). Both enforcement modes are available in either domain: one can introduce explicit inverse tokens with architectural tying, or enforce inverse consistency through relation-alignment penalties. We use architectural tying for braids and involution-alignment for Coxeter generators to demonstrate that Holonomic Networks admit multiple, purely learnable ways to encode the same algebraic structure while keeping inference within a single coherent operator geometry.

Figure 1 and Table 3 (Appendix A) summarize this design language.

## 3.2 Holonomic composition and identity-energy decoding

Let $w = (x_1, \ldots, x_L)$ be an input word over $\mathcal{A}$. The central holonomic computation is the ordered composition of learned token transformations, $P(w)$, defined in (4).

The holonomic template can be viewed as learning an approximate representation map from words to transformations: the assignment $a \mapsto U_a \in SO(d)$ induces a map $\Phi$ by composition, $\Phi(w) = P(w)$. When the training objectives align $\Phi$ with the target relations, $\Phi$ behaves as an approximate homomorphism from the presented group to $SO(d)$, so identity verification becomes a (soft) kernel-membership test. (A more formal phrasing, including the free-monoid and normal-closure viewpoint, is recorded in Appendix C.)

Word identity is read out using a geometric energy relative to the identity transformation, $E(w)$, defined in (5). This scalar energy serves as the task readout. Identity words are expected to produce low energy, while non-identity words should remain separated by a positive margin. The use of a normalized Frobenius energy makes the decoder simple, interpretable, and directly tied to the geometry of the learned representation space.

In the Holonomic Network variable-binding instantiation, the model propagates a latent vector and benefits from per-step renormalization to stabilize long composition. In the present template, the primary object of inference is the composed operator $P(w)$ itself, and the decoder (5) measures its distance to the identity. Accordingly, the algebraic verifiers operate with direct operator composition rather than vector-state transport. This is a decoder-level and representation-level adaptation of the same holonomic principle, not a departure from it.

## 3.3 Task supervision: margin-based energy separation

Let $y(w) \in \{1, 0\}$ denote the supervision label, where $y = 1$ indicates an identity word and $y = 0$ a non-identity word. The shared task-level objective is a margin-based energy separation loss. We fix target margins $m_+$ (for identities) and $m_-$ (for non-identities), with $0 \le m_+ < m_-$, and define

$$\mathcal{L}_{\text{task}}(w, y) = y\left[E(w) - m_+\right]_+ + (1 - y)\left[m_- - E(w)\right]_+, \tag{8}$$

where $[z]_+ = \max(z, 0)$.

This objective does not require a probabilistic classifier over discrete labels; instead, it directly shapes the energy geometry into a low-energy identity basin and a higher-energy non-identity region. For reporting thresholded classification metrics, we use the midpoint threshold

$$\tau = \frac{m_+ + m_-}{2}, \tag{9}$$

and classify a word as identity iff $E(w) < \tau$. This threshold is used only for evaluation and reporting; the training signal is provided by the margin-based energy objective in (8).

### 3.4 Symmetry-aware regularization as geometric alignment

The task loss in (8) is necessary but not sufficient for robust long-length compositional behavior. In the variable-binding setting, the model is trained with a multi-class loss on the transported state, which provides a dense supervisory signal for the token-indexed actions. In contrast, word-identity verification provides only a binary label for an entire word, leaving substantial ambiguity in the geometry induced by the task loss alone. Symmetry-aware alignment and anti-collapse guardrails supply additional geometric information that steers the learned operators toward relation-consistent, non-degenerate regimes. In both the Coxeter and braid settings, we complement it with symmetry-aware regularization terms that encourage the learned generators to align with the algebraic structure of the target domain. The exact relation sets differ by task and are presented in Section 4; here we state the shared template.

Let $\mathcal{R}$ denote a collection of algebraic relation constraints (e.g., braid, commutation, or Coxeter residuals). For each constraint $r \in \mathcal{R}$, let $e_r \geq 0$ denote its squared Frobenius residual. To discourage the optimizer from averaging localized violations, we aggregate these residuals with a temperature-controlled log-sum-exp objective with bias subtraction:

$$\mathcal{L}_{\mathrm{rel}} = \frac{1}{\beta} \log \Big( \sum_{r \in \mathcal{R}} \exp(\beta e_r) \Big) - \frac{\log |\mathcal{R}|}{\beta}, \tag{10}$$

where $\beta > 0$ controls the max-like behavior. The subtraction term normalizes the minimum value to zero and differs from the unnormalized log-sum-exp only by an additive constant.

The resulting quantity $\mathcal{L}_{\mathrm{rel}}$ is the relation-alignment term used in (12).

Continuous optimization can converge to near-trivial solutions (for example, $U_a \approx I$ for all tokens) that reduce relation residuals while eliminating discriminative capacity. We therefore introduce anti-collapse penalties $\mathcal{P}_{\mathrm{anti}}$ alongside relation alignment. Let $\mathcal{P}_{\mathrm{anti}}$ denote auxiliary penalties that discourage degenerate low-order or near-collapsed representations. A representative example is the trace hinge penalty used in both instantiations to discourage collapse toward the identity. Since $\mathrm{Tr}(I) = d$ in $SO(d)$, large traces indicate near-identity operators, and we penalize traces above a margin:

$$\mathcal{L}_{\mathrm{trace}} = \frac{1}{|\mathcal{A}|} \sum_{a \in \mathcal{A}} \max(0, \ \mathrm{Tr}(U_a) - (d - \delta)), \tag{11}$$

with fixed $\delta > 0$. We optimize a weighted objective of the form

$$\mathcal{L} = \lambda_{\mathrm{task}} \mathcal{L}_{\mathrm{task}} + \lambda_{\mathrm{rel}} \mathcal{L}_{\mathrm{rel}} + \sum_{\alpha \in \mathcal{P}_{\mathrm{anti}}} \lambda_\alpha \mathcal{L}_\alpha. \tag{12}$$

As in Section 2.3, these regularizers should be understood as symmetry-aware geometric alignment terms used operationally to stabilize the learned geometry and help preserve energy separation under compositional extrapolation.

Both instantiations use staged optimization in which symmetry alignment is emphasized early and task separation is strengthened later. This training strategy is not part of the holonomic inference mechanism itself, but it is an important practical component of achieving robust separation in the algebraic regimes studied here.

### 3.5 Shared and task-specific components

The unified template described above is intentionally minimal. It isolates a common neural mechanism while leaving task-specific components explicit.

The following components are shared:

- orthogonal generator parameterization via (1),

- holonomic composition via (4),

- identity-energy decoding via (5),

- margin-based task supervision via (8),

- symmetry-aware regularization as geometric alignment via (12).

The following components differ across the two domains:

- tokenization (including explicit inverse tokens in the braid setting),

- algebraic relation sets and associated regularization terms,

- data generation and class-balancing strategy,

- oracle labeling and evaluation protocol (including independent external verification for braid words).

This decomposition is central to the claim of the paper. The shared inference principle remains holonomic and neural, continuous, and composition-based, while the supervision and evaluation machinery is adapted to the algebraic structure and validity requirements of each benchmark.

Evaluation quantities derived from the energy $E(w)$ are defined together with the experimental protocol in Section 5.1. With the shared template in place, the remainder of the methodological description becomes task-specific only in the precise ways listed above. Section 4 therefore presents the Coxeter and braid verifiers not as separate architectures, but as two instantiations of the same holonomic neural principle under different algebraic symmetries and evaluation constraints.

## 4 Two Instantiations of the Same Principle

We now instantiate the unified template of Section 3 in two algebraic settings: Coxeter words for the symmetric group and braid words for the braid group. These two settings represent an escalation in algebraic and verification difficulty: the Coxeter case is finite with involutive generators and an exact internal oracle, whereas the braid case is infinite with explicit inverses and therefore requires stronger guardrails and external oracle auditing. These are presented as two realizations of a common holonomic neural principle, not as unrelated architectures. In both cases, inference is performed by composition of learned orthogonal transformations and decoding by identity energy. What changes are the algebraic relation sets, token conventions, data-generation strategy, and oracle-based evaluation protocol.

From the Holonomic Network perspective, these differences can be viewed as a controlled choice of enforcement modes (architectural tying, loss-based symmetry alignment, and data/oracle hygiene) layered on top of the same holonomic inference computation.

### 4.1 Coxeter instantiation: symmetric-group word identity

We consider the Coxeter presentation of the symmetric group $S_N$ (type $A_{N-1}$), with adjacent transposition generators (Humphreys, 1990)

$$s_1, \ldots, s_{N-1},$$

subject to the standard relations

$$s_i^2 = e, \qquad s_i s_{i+1} s_i = s_{i+1} s_i s_{i+1}, \qquad s_i s_j = s_j s_i \;\; (|i - j| > 1). \tag{13}$$

Here $e$ denotes the group identity, corresponding to id as a permutation and to $I$ under the learned matrix representation. The input is a word over generator indices $i \in \{1, \dots, N-1\}$, and the task is binary identity verification: determine whether the represented permutation is the identity.

A central advantage of the Coxeter setting is that labels can be computed exactly and efficiently in the training pipeline. Given a word $w = (i_1, \dots, i_L)$, we evaluate the induced permutation by applying adjacent swaps to a canonical ordered list. The label is then

$$y(w) = \mathbf{1}\{\pi(w) = \mathrm{id}\}.$$

This exact oracle makes the Coxeter instantiation a clean environment for studying the geometry of the learned verifier and for separating model behavior from label uncertainty.

The Coxeter generator is designed to produce balanced batches while avoiding trivial class cues. At a high level, the pipeline:

1. samples identity-rich candidate words,

2. produces non-identity variants by controlled mutation,

3. applies Coxeter-preserving rewrite scrambling,

4. rechecks labels using the exact oracle, and

5. enforces class balance in the final batch.

This design preserves exact correctness while increasing syntactic diversity of words that represent the same group element. It also reduces the risk that the classifier can exploit superficial token-level regularities rather than the intended algebraic structure.

The Coxeter verifier uses the shared energy-separation objective of Section 3, together with symmetry-aware penalties tailored to (13). The regularization includes:

- **involution penalties** encouraging $U_i^2 \approx I$,

- **braid penalties** encouraging $U_i U_{i+1} U_i \approx U_{i+1} U_i U_{i+1}$,

- **distant commutation penalties** encouraging $U_i U_j \approx U_j U_i$ for $|i - j| > 1$,

- **anti-collapse penalties**, including trace-based and adjacent-pair separation terms that discourage degenerate representations with poor class geometry.

These terms instantiate the generic alignment objective in (12): they bias the learned geometry toward a representation that reflects Coxeter structure and supports stable energy separation.

The Coxeter experiments follow the staged training template of Section 3: an early phase emphasizing structural alignment, followed by stronger task-separation training, and a final polishing phase. The benchmark protocol used in this paper trains on short words ($L \leq 50$) and evaluates on substantially longer lengths up to $L = 5000$. This setting provides the finite-group testbed for the geometric-separation phenomenon analyzed in Section 5.

The Coxeter instantiation serves as the clean geometry case: exact in-pipeline oracle labels, transparent relation structure, and highly interpretable energy histograms and gap statistics. It is the most direct setting in which to visualize symmetry-protected geometric separation.

## 4.2 Braid instantiation: braid-word identity with oracle auditing

We consider braid words in $B_n$, generated by $\sigma_1, \ldots, \sigma_{n-1}$ and their inverses (Birman, 1974). They satisfy braid and distant-commutation relations

$$\sigma_i \sigma_{i+1} \sigma_i = \sigma_{i+1} \sigma_i \sigma_{i+1}, \qquad \sigma_i \sigma_j = \sigma_j \sigma_i \ \ (|i - j| > 1). \tag{14}$$

Unlike the Coxeter setting, generators are not involutions; inverse structure is explicit and is represented in the model through tokenization and orthogonality ($U_{\sigma_i^{-1}} = U_{\sigma_i}^\top$).

The braid-group word problem admits exact algorithmic solutions, including polynomial-time procedures based on Garside normal forms (Garside, 1969). Our goal is not to improve on symbolic solvers, but to use $B_n$ as a stringent non-abelian benchmark for operator-composition generalization and evaluation hygiene. We embed generators into $SO(d)$ with $d = 128$. For comparison, $S_{32}$ has faithful orthogonal representations in dimension 31 (the standard representation) and 32 (permutation matrices). Faithful linear representations of $B_n$ are also known, such as the Lawrence–Krammer representation (Krammer, 2002; Bigelow, 2001), but they are highly structured and not naturally orthogonal. The $SO(128)$ parameterization provides a flexible continuous hypothesis class for satisfying relation constraints while preserving a measurable energy margin.

Braid-word benchmarks are more vulnerable to shortcut exploitation than the Coxeter setting because local motifs and low-cost invariants can correlate with labels in synthetic generators. For this reason, we treat the braid instantiation as a joint model-and-protocol design.

The generator is built to control easy invariants and to avoid tell-tale substrings that appear only under one label. The design includes:

- **relator-based positive construction** with exact-length filling,
- **invariant-matched pure negatives** formed by conjugation $u \, c \, u^{-1}$, where $c$ is a short pure-braid core word and $u^{-1}$ is the formal inverse word (reverse order, invert each token),
- **kernel cancelers in positives** of the form $k \, k^{-1}$, using the same library of short core words, so that core-like motifs occur under both labels,
- **balanced sampling** across classes and lengths.

Concretely, let $\pi_B : B_n \to S_n$ denote the canonical projection sending each generator $\sigma_i$ to the adjacent transposition $(i \ i{+}1)$. We call a word $w$ pure when $\pi_B(w) = \mathrm{id}$, i.e., $w \in \ker(\pi_B)$. We fix a small library $\mathcal{C} \subseteq \ker(\pi_B)$ of short pure-braid words (listed in Appendix B.3). Negatives are generated by sampling $c \in \mathcal{C}$ and conjugating by a random word $u$, which preserves the permutation image and produces nontrivial pure braids in typical cases. Any accidental identities are detected by the external oracle and counted as nonzero `LabelErr`. To prevent a model from using "presence of a core template" as a label cue, positives also include canceling insertions $k \, k^{-1}$ with $k \in \mathcal{C}$; these insertions are identity in $B_n$ but share the same local motifs as the negative cores. This is the main anti-shortcut device used in the braid benchmark and is part of the paper's methodology contribution.

The braid verifier again uses the shared energy-based task objective, but with a different symmetry-alignment bundle. In addition to braid and distant-commutation penalties reflecting (14), the implementation includes anti-collapse and local-geometry penalties tailored to the braid setting (including terms discouraging low-order collapse and promoting adjacent-generator distinctness/compatibility). As in Section 2.3, these terms are used operationally to stabilize the learned geometry and improve separation under extrapolation.

The braid setting is the primary testbed for hygiene-aware evaluation. The model exports a structured evaluation dump (including sequences, generator labels, model predictions, and energy scores), which is then checked by an independent verification script implemented in SageMath (The Sage Developers, 2026). The verifier:

1. reconstructs braid words from dumped token sequences,

2. evaluates identity using a trusted external braid-group oracle,

3. compares oracle labels with generated labels (`LabelErr`),

4. reports accuracy, TPR, and FPR by length.

This pipeline allows us to separate model correctness from generator correctness and to guard against hidden label leakage or construction errors in the synthetic benchmark. In particular, the braid results reported in Section 5 are supported by both internal neural evaluation and independent external verification.

The braid instantiation serves as the oracle-audited case: it demonstrates that the holonomic energy-decoder template remains effective in a structurally richer infinite-group setting, provided that training and evaluation are paired with explicit shortcut resistance and oracle checking.

In particular, the Coxeter instantiation emphasizes extreme length extrapolation under internally generated supervision, whereas the braid instantiation emphasizes shortcut-resistant generation together with independent oracle auditing.

For readers who want a compact side-by-side comparison of the shared holonomic mechanism and the task-specific supervision and evaluation protocols across variable binding, Coxeter, and braid, we provide Table 3 and a brief discussion in Appendix A.

# 5 Results

This section reports the central empirical phenomenon of the paper: symmetry-protected geometric separation under strong length extrapolation, obtained by the holonomic composition mechanism with an identity-energy decoder. We present results for the Coxeter and braid instantiations introduced in Section 4, together with hygiene and robustness checks designed to reduce shortcut explanations. Throughout, evaluation of a word consists only of composing learned orthogonal transformations and computing the energy-to-identity score in (5).

## 5.1 Experimental protocol and evaluation quantities

In both instantiations we train on short words, with maximum training length $L_{\max}^{\mathrm{train}} = 50$, and evaluate on lengths up to $L = 5000$. All reported metrics are computed at fixed lengths and are intended to probe extrapolation beyond the training regime, rather than interpolation within it.

For each word $w$, the model produces an identity energy $E(w)$ defined in (5). We report both thresholded classification metrics (TPR/FPR/accuracy) using the midpoint threshold $\tau$ in (9), and geometry-aware statistics that directly characterize separation:

$$E_{\max}^{\mathrm{id}} = \max_{w \in \mathcal{D}_{\mathrm{id}}} E(w), \qquad E_{\min}^{\mathrm{non}} = \min_{w \in \mathcal{D}_{\mathrm{non}}} E(w), \qquad \Delta = E_{\min}^{\mathrm{non}} - E_{\max}^{\mathrm{id}}.$$

A positive empirical gap $\Delta > 0$ certifies separation on the evaluated sample.

## 5.2 Coxeter: stable geometric separation to length 5000

We first report results for the Coxeter verifier on the Coxeter presentation of $S_{32}$. This setting provides a clean finite-group testbed with exact oracle labels computed by permutation execution (Section 4.1).

Figure 2 visualizes the energy distributions of identity and non-identity words at length $L = 5000$ on a log scale. Identity words concentrate at energies several orders of magnitude smaller than non-identity words, with an empty region separating the two supports. This plot directly exhibits the phenomenon we call symmetry-protected geometric separation: the identity set forms a low-energy basin near the neutral element, while non-identity words remain well separated in energy.

Figure 3 reports the max-identity energy $E_{\max}^{\mathrm{id}}$ and min-non-identity energy $E_{\min}^{\mathrm{non}}$ as a function of length. While $E_{\max}^{\mathrm{id}}$ increases mildly with length (consistent with accumulation under repeated composition), it

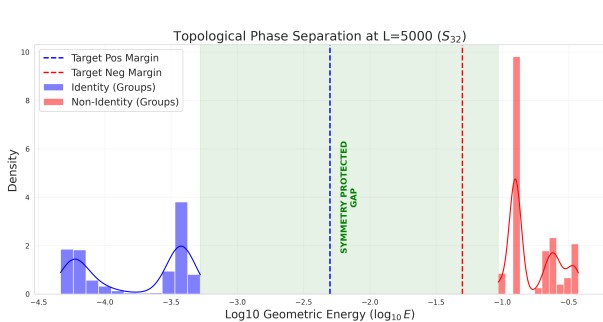

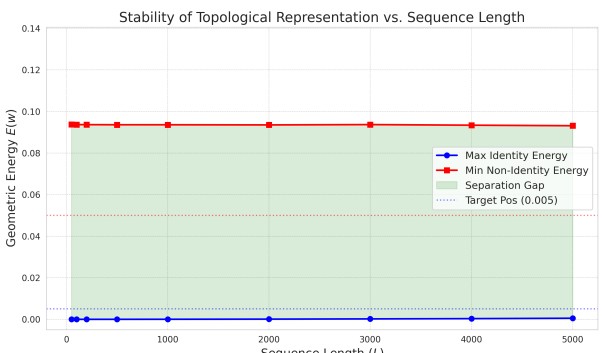

Figure 2: **Coxeter phase separation at** $L = 5000$ **in** $S_{32}$**.** Log-energy distributions for identity and non-identity words at length $L = 5000$. The two classes occupy sharply separated energy regimes, with an empty gap region between them.

Figure 3: **Coxeter stability of geometric separation vs. length.** Maximum identity energy and minimum non-identity energy as a function of word length. The shaded region indicates the empirical separation gap, which remains strongly positive under length extrapolation to $L = 5000$.

| Length $L$ | TPR | FPR | $E_{\max}^{\mathrm{id}}$ | $E_{\min}^{\mathrm{non}}$ | Gap $\Delta$ |
|---|---|---|---|---|---|
| 50 | **1.000** | **0.000** | 7.551e-07 | 9.40e-2 | 9.40e-2 |
| 100 | **1.000** | **0.000** | 1.389e-06 | 9.40e-2 | 9.40e-2 |
| 200 | **1.000** | **0.000** | 3.070e-06 | 9.40e-2 | 9.40e-2 |
| 500 | **1.000** | **0.000** | 1.105e-05 | 9.40e-2 | 9.40e-2 |
| 1000 | **1.000** | **0.000** | 3.317e-05 | 9.40e-2 | 9.40e-2 |
| 2000 | **1.000** | **0.000** | 9.899e-05 | 9.40e-2 | 9.39e-2 |
| 3000 | **1.000** | **0.000** | 2.039e-04 | 9.40e-2 | 9.39e-2 |
| 4000 | **1.000** | **0.000** | 3.390e-04 | 9.39e-2 | 9.39e-2 |
| 5000 | **1.000** | **0.000** | 5.255e-04 | 9.39e-2 | 9.39e-2 |

Table 1: **Coxeter identity verification in** $S_{32}$**:** generalization under length extrapolation. The model is trained on $L \leq 50$ and evaluated up to $L = 5000$. In addition to perfect sampled TPR/FPR, the table reports explicit geometric separation via $E_{\max}^{\mathrm{id}}$, $E_{\min}^{\mathrm{non}}$, and the gap $\Delta$.

remains far below $E_{\min}^{\mathrm{non}}$, and the empirical gap $\Delta$ remains positive and large across all evaluated lengths up to $L = 5000$.

Table 1 reports thresholded metrics together with the geometry-aware separation statistics. On the reported evaluation protocol, the model maintains perfect sampled separation (TPR = 1, FPR = 0) and a persistently positive gap $\Delta$ from $L = 50$ through $L = 5000$. Importantly, this table couples classification performance to explicit geometric quantities, making the claim of separation auditable beyond a single threshold choice.

### 5.3 Braids: oracle-validated extrapolation in $B_8$

We next report results for braid-word identity verification in $B_8$. This setting is structurally richer and more vulnerable to generator artifacts, and therefore we treat the evaluation protocol itself as part of the scientific contribution.

To guard against label leakage or silent errors in synthetic generation, we evaluate using a dump-and-verify protocol: the neural model exports per-example predictions and energies, and an independent SageMath oracle (The Sage Developers, 2026) recomputes braid identity and reports accuracy, TPR, FPR, and label mismatch (`LabelErr`) by length. This yields a strong separation between (i) the neural inference mechanism, which remains purely geometric, and (ii) evaluation validity, which is ensured by external oracle verification.

| Length $L$ | Count | Acc | TPR | FPR | LabelErr |
|---|---|---|---|---|---|
| 50 | 2000 | 1.0000 | 1.0000 | 0.0000 | 0.0000 |
| 100 | 2000 | 1.0000 | 1.0000 | 0.0000 | 0.0000 |
| 200 | 2000 | 1.0000 | 1.0000 | 0.0000 | 0.0000 |
| 500 | 2000 | 1.0000 | 1.0000 | 0.0000 | 0.0000 |
| 1000 | 2000 | 1.0000 | 1.0000 | 0.0000 | 0.0000 |
| 2000 | 2000 | 1.0000 | 1.0000 | 0.0000 | 0.0000 |
| 5000 | 2000 | 1.0000 | 1.0000 | 0.0000 | 0.0000 |

Table 2: **Braid identity verification in $B_8$: SageMath oracle report on an evaluation dump.** The dump contains 14,000 examples (7 lengths $\times$ 2000 examples per length). The independent oracle verifies perfect sampled agreement and reports zero detected generator-label mismatches.

Table 2 gives the SageMath oracle report on an evaluation dump comprising 7 lengths $\times$ 2000 examples per length (total 14,000 examples). Across lengths up to $L = 5000$, the report shows perfect sampled agreement with the oracle (100% accuracy, TPR $= 1$, FPR $= 0$) and zero detected label mismatches (LabelErr$= 0$).

These braid results support the central claim of the paper: the holonomic geometric verifier agrees with an independent external oracle on a large evaluation dump under length extrapolation to $L = 5000$. As with the Coxeter setting, this does not constitute a theorem-level solution to the braid word problem. It does, however, provide clean empirical evidence that the holonomic inference mechanism, coupled with symmetry-aware training and hygiene-aware evaluation, can maintain correctness far beyond the training-length regime.

Because the headline metrics are strong, we present targeted ablations next to demonstrate necessity of the structural and hygiene components, followed by evaluation hygiene checks that address shortcut and label-leakage concerns.

### 5.4 Ablations: necessity of structural and hygiene components

The separation results rely on two ingredients: symmetry-aware structural alignment and shortcut-resistant evaluation. We tested this dependence with targeted knock-out experiments in which individual components were removed while keeping the remainder of the training protocol fixed. In each case, performance on short lengths remained strong, but long-horizon separation degraded markedly.

- **Relation-penalty aggregation.** Replacing the log-sum-exp aggregation in (10) used for relation residuals with a simple mean-squared residual encourages the optimizer to average localized violations rather than suppress worst-case violations. Under this change, the learned operators continue to fit the short-length training regime, while the long-length energy gap narrows and error rates increase shortly beyond this regime.

- **Anti-collapse guardrails.** Removing the trace hinge and adjacent-pair separation terms leads to degenerate solutions in which generators drift toward near-identity or low-order behavior. In the Coxeter setting this manifests as a collapse toward representations that satisfy relations while losing discriminative capacity; in the braid setting it increases false positives on long words.

- **Data hygiene.** Removing kernel cancelers from the braid positive construction reintroduces motif-based shortcuts. Training converges rapidly, but long-length behavior degrades: the model becomes sensitive to kernel-like substrings that appear frequently in long words, and the true positive rate on long identities drops sharply.

These ablations support the interpretation of the main results as a consequence of structured alignment and hygiene-aware evaluation rather than of superficial statistics.

### 5.5 Evaluation hygiene and robustness checks

The empirical strength of long-length generalization results depends critically on ruling out shortcut explanations. We therefore emphasize the following safeguards, which are integrated into the experimental protocol.

Coxeter labels are computed by direct permutation execution, eliminating label noise and making it possible to apply rewrite scrambling followed by revalidation. This ensures that the reported separation is not an artifact of approximate labeling.

Braid evaluations are accompanied by an explicit external oracle verification pipeline that reports both prediction metrics and label mismatch. The `LabelErr` metric is a direct safeguard against generator errors and self-confirmation bias: a perfect classifier would be uninformative if its evaluation labels were wrong. The reported `LabelErr`=0 across all lengths in Table 2 therefore strengthens the credibility of the extrapolation claim.

The results in this section establish a robust empirical phenomenon: under symmetry-aware training, identity and non-identity words occupy separated energy regimes that remain stable under large length extrapolation on the reported benchmark distributions. They do not establish correctness for all words, nor do they prove that the learned generators form an exact faithful representation of the underlying groups.

## 6 Discussion

The results in Section 5 show that holonomic operator composition, paired with symmetry-aware alignment and hygiene-aware evaluation, can sustain a persistent energy margin under substantial length extrapolation in both finite (Coxeter) and infinite (braid) non-abelian settings. The braid claims are supported by an independent SageMath audit of exported evaluation dumps. This section interprets the observed geometric separation, discusses validity considerations for synthetic algebraic benchmarks, and outlines extensions of the operator-and-verifier template to broader rewrite-invariant settings.

### 6.1 Interpreting "symmetry-protected geometric separation"

The results sharpen a familiar concern in long-horizon sequence modeling: continuous representations can drift under deep composition. Here the identity-energy decoder exposes this behavior as an explicit, auditable margin. A persistent gap between identity and non-identity energies indicates that the learned operator geometry remains stable under substantially increased compositional depth.

The symmetry-aware objectives play a specific role in this picture. Relation alignment steers the learned actions toward a representation in which semantics-preserving rewrites act as near-invariances, while anti-collapse guardrails exclude degenerate solutions (such as near-identity actions) that can reduce relation residuals without supporting discrimination. The ablations in Section 5.4 show that removing these components degrades long-length behavior even when short-length accuracy remains high.

In the sense defined in Section 2.3, symmetry-protected geometric separation is a measurable property of the learned energy landscape that can be stress-tested under generator shift, alternative rewrite families, and independent oracle audits.

### 6.2 Validity considerations and evaluation protocols

Strong extrapolation results in synthetic algebraic benchmarks can be undermined by shortcuts. The paper addresses this risk directly.

- **Generator artifacts.** Synthetic data can leak class information via motifs, length distributions, or easy invariants. For this reason, we emphasize balanced construction and, in the braid case, explicit anti-shortcut design (invariant-matched negatives and kernel cancelers).

- **Label leakage and self-confirmation.** If evaluation labels are produced by the same generator that produced training labels, errors can silently propagate. The braid dump-and-verify protocol addresses this by recomputing identity using an independent SageMath oracle and explicitly reporting label mismatch rates.

- **Threshold dependence.** Any scalar score can be made to look good at a particular threshold. We therefore report geometry-aware separation statistics (not only thresholded accuracy), so that the phenomenon is visible as a distributional gap rather than a single operating point.

These design choices are part of the scientific content of the paper: the goal is not merely to report high accuracy, but to support a credible interpretation that the network is learning algebraically meaningful structure rather than exploiting artifacts.

### 6.3 Opportunities and extensions

The two instantiations in this paper are deliberately narrow: they are intended to expose a general design pattern rather than to exhaust the space of applications. From a broader viewpoint, Holonomic Networks can be understood as a framework for learning composable operators in a structured hypothesis class and attaching auditable verifiers to their compositions. In the present work the operators live in $SO(d)$ and the verifier is identity energy, but neither choice is essential.

This operator-and-verifier view also suggests a natural interface with geometric deep learning beyond static data: if symmetries vary across time or context, then holonomic composition provides a principled way to accumulate local symmetry information into a global certificate. Connections to gauge-equivariant constructions (Cohen et al., 2019; Bronstein et al., 2021) and to flow-equivariant recurrent models (Keller, 2025) offer concrete starting points for bringing these ideas to broader sequential domains.

Token-indexed actions $U_x$ provide a learned operator basis whose semantics is expressed by composition. This connects naturally to operator-learning perspectives in machine learning, including neural operator models for map-to-map prediction and dynamical systems (Lu et al., 2021; Kovachki et al., 2023). A direct extension is to broaden the operator family beyond $SO(d)$ to other Lie groups and structured transformation classes (e.g., unitary, symplectic, or rigid-motion groups), while preserving the stability properties of near-isometric composition emphasized in the original Holonomic Network formulation and its relation to long-horizon drift in recurrent models (Sung, 2026; Arjovsky et al., 2016; Wisdom et al., 2016).

The identity-energy decoder used here is a special case of a more general principle: interpretability and auditability improve when the decoder produces a scalar certificate derived from the composed operator. Beyond identity testing, analogous geometric verifiers can target membership in other algebraic subsets, distances to constraint manifolds, or multi-class predicates defined by equivalence relations.

Beyond group-word identity, rewrite-invariant computation is a recurring theme across mathematics and computer science: term-rewriting systems and completion procedures formalize equivalence by rewrite closure (Baader & Nipkow, 1998; Terese, 2003; Knuth & Bendix, 1970), and related invariance questions arise in symbolic algebra and algorithmic group theory (Epstein et al., 1992; Holt et al., 2005). The Holonomic Network template suggests a neural route to such semantics-preserving computation: learn structured actions whose holonomic composition admits simple, auditable readouts that are stable under large families of semantics-preserving rewrites, paralleling the role of invariance and equivariance in geometric deep learning (Cohen & Welling, 2016; Bronstein et al., 2021; Kondor & Trivedi, 2018).

A complementary direction is certification. For finite-state or finite-group regimes, one can export learned operators, verify defining relations to a specified tolerance, and bound the separation margin needed for a fixed decision rule. In the Coxeter setting, this suggests projecting to or recovering an exact discrete representation when possible, and then certifying a positive minimum distance from the identity over the finite image. For broader rewrite systems, one can couple holonomic readouts with trusted checkers and validated numerics, turning the learned model into a proposal mechanism whose outputs admit machine-checkable certificates. Such a workflow is relevant whenever correctness matters in long compositions, including symmetry-preserving simulation and quantum circuit equivalence in physics, composition of constrained

transformations in biological dynamical models, and audit-and-verify pipelines for protocol traces and policy compliance in cybersecurity.

The modularity highlighted in this paper suggests several directions:

- **Operator families beyond** $SO(d)$**.** Extend the operator hypothesis class to alternative Lie groups and structured transformations, or to continuous-time flows, and study how stability and separation depend on geometry and parameterization.

- **Verifiers beyond identity.** Replace identity energy with certificate-like readouts for other algebraic or semantic predicates (e.g., distances to subsets, conjugacy-sensitive observables, or multi-class equivalence tests).

- **Rewrite-invariant learning beyond groups.** Apply the same holonomic operator-and-verifier template to domains where semantics is defined by rewrite systems (program transformations, compiler equivalence, symbolic algebra), using trusted external checkers to audit correctness when available.

- **Auditing as a reusable evaluation template.** Generalize the dump-and-verify methodology: export model decisions and certificates, and validate them against independent oracles or checkers to separate model correctness from dataset construction.

## 7 Conclusion

Holonomic Networks implement computation by composing token-indexed geometric actions. The variable-binding instantiation shows that a standard neural training objective can learn such actions and generalize by the associativity and closure of the induced composition, without requiring symbolic execution in the forward computation.

This paper shows that the same operator-composition engine supports a different decoder regime. Replacing variable readout by an identity-energy decoder yields a unified template for algebraic word identity verification in which generators are learned as orthogonal transformations in $SO(d)$, words are evaluated by ordered composition, and identity is read out by a geometric energy relative to the neutral element. When coupled with symmetry-aware alignment objectives, anti-collapse guardrails, and hygiene-aware evaluation protocols, this template exhibits stable geometric separation under substantial length extrapolation.

Our contribution is methodological and foundational: we articulate a reusable operator-and-verifier template and pair it with auditable evaluation practices showing that continuous geometric representations, when properly constrained, can support discrete-algebraic semantics over extreme compositional depth. In both Coxeter and braid instantiations, holonomic operator composition produces a stable geometric certificate for identity that generalizes far beyond the training regime, and the braid results are independently validated by an external SageMath oracle. These findings suggest that the gap often perceived between continuous neural representations and exact discrete reasoning is not solely a matter of model capacity, but frequently a consequence of unconstrained geometry and weak supervisory signals; enforcing structure and auditing claims can shift that boundary substantially in practice.

### Reproducibility Statement

All experiments reported in this paper are fully specified by the model definition (orthogonal generator parameterization via skew-symmetric matrix exponentials), the identity-energy decoder, the stated symmetry-aware regularization objectives, and the described data-generation and oracle-validation protocols for each domain. In particular, the braid evaluation includes an external oracle audit pipeline that recomputes identity using SageMath on exported per-example prediction dumps, enabling independent verification of both predictive metrics and generator-label consistency. We will release the complete code and scripts required to reproduce training, evaluation, figure generation, and oracle verification.

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

## A    Comparison across variable binding, Coxeter, and braid

Table 3 summarizes the conceptual relation among (i) the Holonomic Network variable-binding task, (ii) the Coxeter word-identity verifier, and (iii) the braid word-identity verifier. The table is organized around the distinction emphasized throughout this paper: shared holonomic inference mechanism versus task-specific supervision and evaluation.

The Coxeter and braid models are best viewed as two instantiations of a single design pattern introduced by Holonomic Network: learn a geometric action for each token, compose actions holonomically, and decode by a simple task-aligned criterion. The difference is not in the existence of multiple architectures, but in how the same compositional neural principle is specialized to distinct algebraic symmetries and validated under different levels of oracle control. This distinction helps interpret the empirical results and robustness analyses in Section 5.

## B    Implementation and Evaluation Details

This appendix records the minimal implementation and evaluation details required to reproduce the claims of the paper without relying on extensive supplemental material. All experiments share the same holonomic inference template: learned orthogonal token actions, ordered composition, and a simple task-aligned decoder. Oracle computation is used only for dataset label generation and for external verification of evaluation dumps in the braid setting.

### B.1    Variable binding benchmark

To reproduce the results of the Holonomic Network variable-binding task in (Sung, 2026), we follow the experimental setup detailed below. The variable-binding benchmark samples programs consisting of swap

| Component | Original Holonomic Network (variable binding) | Coxeter verifier | Braid verifier |
|---|---|---|---|
| Task output | Value at distinguished slot after swap program | Identity / non-identity of Coxeter word | Identity / non-identity of braid word |
| Core composition mechanism | Holonomic composition of learned token actions on latent state | Holonomic composition of learned generator operators | Holonomic composition of learned generator/inverse operators |
| Decoder | Neural classifier readout on transported latent state | Energy-to-identity $(\lVert P(w) - I \rVert_F^2 /d)$ | Energy-to-identity $(\lVert P(w) - I \rVert_F^2 /d)$ |
| Structural enforcement mode | $SO(d)$ actions + state normalization | $SO(d)$ actions + relation losses + symmetry-preserving rewrites | $SO(d)$ actions + inverse transpose-tying + relation losses + oracle-audited evaluation |
| Symmetry-aware training structure | Implicit compositional inductive bias through holonomic transport | Coxeter relation penalties + anti-collapse regularization | Braid/commutation penalties + anti-collapse/local-geometry regularization |
| Label / oracle pipeline | Supervision from generated swap-execution labels | Exact in-pipeline oracle via permutation execution | Generator labels + independent SageMath oracle audit |
| Primary evaluation risk | Short-length shortcut fitting in sequence modeling | Syntactic shortcut cues in generated words | Generator artifacts / invariant leakage / label mismatch |
| Primary safeguard emphasized here | Holonomic compositional inductive bias | Exact oracle + rewrite scrambling + balance | Anti-shortcut generation + dump-and-verify external oracle |

Table 3: Conceptual comparison of the Holonomic Network variable-binding setting and the two algebraic word-identity instantiations studied in this paper. The shared mechanism is holonomic neural composition; the decoder and evaluation protocols are task-specific.

operations applied to an initial array of values $(0, 1, \ldots, N-1)$. The supervision target is the value occupying a distinguished slot (slot 0) after symbolic execution of the sampled swap program.

A batch consists of:

1. an initial state $s_0 \in V^N$ with $s_0[k] = k$,

2. a program $w = (x_1, \ldots, x_L)$, where each $x_t$ is an index into a fixed list of swap pairs,

3. the label $y = s_L[0]$, where $s_L$ is obtained by applying swaps sequentially.

The Holonomic Network model for variable binding learns:

- value embeddings $\{e_v\}_{v \in V} \subset \mathbb{R}^d$,

- token actions $U_a \in SO(d)$ via the exponential of skew-symmetric matrices,

- a linear head for classification over values.

Inference applies holonomic latent transport $h_t = \mathcal{N}(U_{x_t} h_{t-1})$ followed by a standard classifier readout. The per-step renormalization $\mathcal{N}$ is a geometric stabilization on the unit sphere (implemented with a $10^{-16}$ machine-epsilon buffer to prevent division-by-zero and gradient singularities in `float64`).

Training uses a moving-target curriculum that gradually increases the maximum sampled program length until reaching $L_{\max}^{\text{train}} = 50$, while evaluation probes extrapolation up to length $L = 5000$.

## B.2 Coxeter instantiation: data generator, oracle, and structural losses

For Coxeter words in the symmetric group $S_N$ (type $A_{N-1}$), labels are computed by executing the induced adjacent swaps on permutations. Given a word $w$, the label is $y(w) = 1$ iff the resulting permutation equals the identity. This oracle is exact and is used in both data generation (to revalidate scrambled candidates) and evaluation.

The Coxeter batch generator constructs identity-rich candidates and non-identity variants, applies Coxeter-preserving rewrite scrambling (commutations and braid rewrites), then rechecks identity via the exact oracle before admitting examples. The final batch is enforced to be class-balanced and shuffled.

The Coxeter verifier uses the identity-energy decoder:

$$E(w) = \frac{1}{d}\|P(w) - I\|_F^2, \qquad P(w) = U_{x_L} \cdots U_{x_1},$$

where $U_i \in SO(d)$ are learned generator matrices. Identity prediction is made by thresholding $E(w)$ at the midpoint of the target margins.

The symmetry-aware objective includes relation-alignment penalties corresponding to the Coxeter relations:

- involution: $U_i^2 \approx I$,

- braid: $U_i U_{i+1} U_i \approx U_{i+1} U_i U_{i+1}$,

- distant commutation: $U_i U_j \approx U_j U_i$ for $|i - j| > 1$,

together with anti-collapse terms that discourage degenerate representations. Relation residuals are aggregated using a log-sum-exp construction with bias subtraction to reduce dependence on the number of constraints.

## B.3 Braid instantiation: generator design and oracle auditing

The braid instantiation uses a vocabulary containing both positive and inverse tokens. Positive generators are parameterized in $SO(d)$, and inverse tokens are realized geometrically by transposition: $U_{\sigma_i^{-1}} = U_{\sigma_i}^\top$. This preserves a single coherent orthogonal geometry while representing inverses by transposition.

The braid generator is designed to reduce trivial class cues:

- identity words are constructed by inserting sampled relators until reaching an exact target length,

- non-identity words are constructed as invariant-matched "pure negatives" of the form $u \cdot \text{core} \cdot u^{-1}$, followed by relator filling to exact length,

- kernel cancelers are inserted into positives to prevent motif-based discrimination.

All examples are produced at exact even lengths, and batches are balanced.

**Core library $\mathcal{C}$ used in the generator.** In our scripts, $\mathcal{C}$ is instantiated as the union of two short pure-braid template families:

- **type-1 cores:** $c = \sigma_i^2 \sigma_j^{-2}$ for $i \neq j$,

- **type-2 cores:** $c = \sigma_i^2 \sigma_{i+1}^2 \sigma_i^{-2} \sigma_{i+1}^{-2}$ for $i = 1, \ldots, n-2$.

Both families lie in $\ker(\pi_B)$ (their permutation image is id) and have balanced signed exponent sum. Negatives are generated from $u\,c\,u^{-1}$ and then filled to the target length by inserting additional identity relators; positives are generated by relator insertions and additionally include kernel cancelers $k\,k^{-1}$ with $k \in \mathcal{C}$.

| Setting | Variable binding (Holonomic Network) | Coxeter verifier | Braid verifier |
|---|---|---|---|
| Group / task domain | swap programs ($N{=}10$) | $S_{32}$ (Coxeter $A_{31}$) | $B_8$ (braid group) |
| Representation dim $d$ | 32 | 128 | 128 |
| Token actions | $SO(d)$, exp(skew) | $SO(d)$, exp(skew) | $SO(d)$, exp(skew) |
| Inverse handling | n/a | involution-aligned | $U_{a^{-1}} := U_a^\top$ |
| Decoder | value classifier | identity energy $\|P-I\|_F^2/d$ | identity energy $\|P-I\|_F^2/d$ |
| Target margins $m_+, m_-$ | n/a | 0.005, 0.05 | 0.01, 1.0 |
| Training max length | 50 | 50 | 50 |
| Evaluation max length | 5000 | 5000 | 5000 |
| Batch size | 64 | 128 | 128 |
| Total steps | 60000 | 60000 | 80000 |
| Primary oracle | swap execution (labels) | permutation execution (labels) | SageMath audit (evaluation) |

Table 4: Primary configuration summary for the three experimental components of the paper.

The braid verifier again uses identity-energy decoding, together with alignment penalties for the braid and commutation relations, and additional anti-collapse/local-geometry terms (e.g., short-order penalties and adjacent commutator/distinctness guardrails). These terms are used as empirical stabilizers of the learned geometry; they are not claimed as theorem-level enforcement.

To separate predictive correctness from generator correctness, evaluation proceeds via an explicit dump-and-verify protocol:

1. the model exports a JSONL dump containing, per example, $(L, \texttt{seq}, y_{\text{gen}}, \hat{y}, E(w))$,

2. an independent SageMath script reads the dump, performs free reduction for speed, computes the oracle identity label $y_{\text{oracle}}$, and reports accuracy/TPR/FPR by length,

3. the script also reports `LabelErr`, the fraction of examples for which $y_{\text{gen}} \neq y_{\text{oracle}}$.

The reported braid results in Section 5 are supported by this independent oracle audit.

### B.4 Hyperparameter summary

Table 4 summarizes the primary configuration choices for the three components of the paper. Unless otherwise stated, computations use double precision (`float64`) for numerical stability under long composition, and training follows a staged schedule in which structural alignment is emphasized early and task separation is strengthened later.

## C Supplementary Expository Material

This appendix provides supplementary exposition referenced from the main text: (i) a self-contained restatement of the unified template and identity-energy decoder, and (ii) a slightly more formal algebraic interpretation of word identity as an approximate kernel-membership test.

### C.1 Self-contained restatement of the unified template

For convenience, we restate the standard $SO(d)$ parameterization and the identity-energy decoder in one place.

As in the original Holonomic Network construction, each $U_a$ is parameterized by exponentiating a skew-symmetric matrix:

$$A_a = M_a - M_a^\top, \qquad U_a = \exp(A_a) \in SO(d), \tag{15}$$

where $M_a \in \mathbb{R}^{d \times d}$ is unconstrained. This parameterization provides a smooth, trainable family of orthogonal transformations while preserving a strong geometric inductive bias. In particular, composition of tokens remains in the same structured transformation class, which is essential for stable long-horizon reasoning.

Let $w = (x_1, \ldots, x_L)$ be an input word over $\mathcal{A}$. The holonomic computation is the ordered composition of learned token transformations:

$$P(w) = U_{x_L} \cdots U_{x_1}. \tag{16}$$

The holonomic template can be viewed as learning an approximate representation map from words to transformations. Let $\mathcal{A}^*$ denote the free monoid on the token set $\mathcal{A}$. The assignment $a \mapsto U_a \in SO(d)$ induces a map $\Phi : \mathcal{A}^* \to SO(d)$ by composition, $\Phi(w) = P(w)$. When the training objectives align $\Phi$ with the target relations, $\Phi$ behaves as an approximate homomorphism from the presented group to $SO(d)$. Identity verification then becomes a (soft) kernel membership test: determine whether $w$ lies in the normal closure of the defining relations by checking whether $\Phi(w)$ is close to the identity. The energy $E(w)$ in (17) provides a continuous proxy for distance to the kernel and makes semantic invariance under relation-preserving rewrites experimentally visible as geometric separation. We emphasize that this is the algebraic analogue of the original Holonomic Network latent transport mechanism: reasoning is performed by composition of learned geometric actions. The difference from the variable-binding setting is that we now decode directly from the composed operator $P(w)$, rather than from a transported probe vector.

Word identity can then be read out using a geometric energy relative to the identity transformation:

$$E(w) = \frac{1}{d} \| P(w) - I \|_F^2 = 2 - \frac{2}{d} \operatorname{Tr}(P(w)). \tag{17}$$

## C.2 Evaluation quantities used throughout the paper

To make the subsequent results directly comparable across tasks, we summarize the common evaluation quantities derived from the energy $E(w)$:

- **Thresholded metrics:** accuracy, true positive rate (TPR), and false positive rate (FPR), using the threshold $\tau$ in (9).

- **Geometric separation statistics:**

$$E_{\max}^{\mathrm{id}} = \max_{w \in \mathcal{D}_{\mathrm{id}}} E(w), \qquad E_{\min}^{\mathrm{non}} = \min_{w \in \mathcal{D}_{\mathrm{non}}} E(w),$$

and the empirical gap

$$\Delta = E_{\min}^{\mathrm{non}} - E_{\max}^{\mathrm{id}}.$$

A positive $\Delta$ provides a direct certificate of separation on the evaluated sample.

These quantities realize the operational notion introduced in Section 2: symmetry-protected geometric separation is observed when symmetry-aware training produces a stable positive gap $\Delta$ and low error rates under substantial length extrapolation.

With the shared template in place, the remainder of the methodological description becomes task-specific only in the precise ways listed above. Section 4 therefore presents the Coxeter and braid verifiers not as separate architectures, but as two instantiations of the same holonomic neural principle under different algebraic symmetries and evaluation constraints.

