# OpenReview forum: "Symmetry-Protected Geometric Separation for Verified Algebraic Reasoning"
_TMLR — Rejected by TMLR_

### Review · Reviewer_JXKu · 2026-04-05

**Summary Of Contributions:**

The paper claims that Holonomic Networks, which implement sequence computation by ordered composition of learned orthogonal token actions in SO(d), can be redirected from variable-binding to algebraic word identity verification by substituting the classifier decoder with a normalized Frobenius-distance-to-identity energy score. Two instantiations are reported, for finite Coxeter words in S_32 and infinite braid words in B_8, yielding perfect sampled classification under 100x length extrapolation. The architectural contribution amounts to a decoder substitution applied to an arXiv preprint that constitutes the entire inference foundation; no explicit comparative baselines are presented; and all empirical claims rest on single-run evaluations carrying no variance estimates, no confidence intervals, and no significance tests.

**Audience:**

No

**Audience Explanation:**

The absence of comparative baselines means the reader cannot determine whether the length extrapolation phenomenon is distinctive to the holonomic parameterization or reproducible with a transformer or multiplicative recurrent network trained under equivalent conditions. The absence of variance estimates means the reported perfect classification figures cannot be taken as reliable point estimates. If the authors provide the comparative and statistical evidence described in the requested changes, the core finding would be publishable and of genuine interest to the TMLR community. In the current form, the findings are suggestive but insufficiently evidenced for the audience to draw warranted conclusions from them.

**Claims And Evidence:**

No

**Claims Explanation:**

The complete inference mechanism of this paper is SO(d) parameterization via matrix exponentials of skew-symmetric matrices, holonomic operator composition, per-step renormalization, and the parallel-transport interpretation, which originates in Sung (2026) [1]. The claimed contribution of the submission is the observation that replacing the variable readout of [1] with a Frobenius energy decoder repurposes the same composition engine for word identity. This is a decoder-level substitution that leaves every component of the trained model intact; describing it as a distinct methodological template overstates its independence from [1].

The use of norm-preserving multiplicative recurrent updates for compositional stability has substantial prior art. Arjovsky, Shah, and Bengio [2] introduced unitary evolution recurrent neural networks at ICML 2016, and Wisdom et al. [3] simultaneously proposed full-capacity unitary recurrent networks, both exploiting bounded-spectrum operator products for the same stability purpose invoked here. The idea that neural networks solving group-structured tasks internally learn approximate group representations is well-established in the mechanistic interpretability literature: Nanda et al. [4] demonstrated at ICLR 2023 that transformers trained on modular arithmetic develop Fourier-basis representations of the cyclic group, a phenomenon conceptually equivalent to the energy separation claimed here. The grokking benchmark [5], which is not cited, establishes that algebraic structure emerges reliably from supervised training on finite-group tasks, including under length or compositional generalization pressure. The submission does not compare against any of these findings or architectures, making it impossible to assess whether the holonomic parameterization provides a genuine advantage over simpler baselines.

Within the symbolic reasoning literature, Lample and Charton [6] showed that transformers learn to solve formal mathematical problems including symbolic integration, and Charton [7] showed that transformers master linear algebraic operations with near-perfect accuracy. Deletang et al. [8] at ICLR 2023 systematically characterized the representational capacity of recurrent and attention models on Chomsky-class tasks, a framework that directly subsumes finite-group word problems. Anil et al. [9] studied length generalization in large language models under extrapolation protocols structurally similar to the present benchmark. None of these directly competing results are presented in comparison. The paper's dismissal of standard sequence models as exhibiting "severe drift" is asserted rather than demonstrated empirically, and no transformer or recurrent baseline trained on the same data under identical conditions appears anywhere in the manuscript.

The terminology of "symmetry-protected geometric separation" borrows from condensed matter physics in a manner that is acknowledged by the authors themselves in Section 2.3 to be purely empirical and descriptive rather than theorem-level: they state that Sung's (2026) analysis provides formal guarantees "under a realizability condition" whereas the present framing "summarizes an empirical separation phenomenon." Presenting a distributional energy gap as "protection" without a supporting formal guarantee overstates the theoretical content of the contribution.

Tables 1 and 2 report uniform point estimates of TPR = 1.0000 and FPR = 0.0000 across all evaluated lengths. No variance over random training seeds is reported anywhere in the paper. No confidence intervals accompany the energy gap statistics. No information is provided on the number of independent training runs, or whether the reported model corresponds to the best, median, or a representative run. For the Coxeter evaluation, the number of words sampled per length is not stated in the main text; without this, even a binomial confidence interval for perfect classification cannot be computed by the reader. For the braid evaluation, 2000 examples per length permit a nominal 95 percent confidence interval of approximately [0.9982, 1.0000] for perfect classification, but this is neither reported nor discussed. The energy gap Delta = 9.39 to 9.40e-2 is presented as a robust certificate, yet its variance across seeds and its sensitivity to the target margin hyperparameters m+ and m- are not examined. The classification threshold tau in equation (9) is the midpoint of the two target margins, which are themselves tunable hyperparameters; the reported TPR/FPR figures depend directly on this pre-specified operating point, and no receiver-operating-characteristic analysis is included.

No ablation over the SO(d) representation dimension d is provided. The Coxeter verifier uses d = 32, citing the standard representation dimension of S_32, while the braid verifier uses d = 128 without systematic justification. No sensitivity analysis over the lambda_task, lambda_rel, and anti-collapse penalty weights, which appear in equation (12), is presented. The staged training curriculum is described qualitatively, but the number of steps per stage and the criteria governing stage transitions are not reported in a form sufficient for reproduction. The reproducibility statement promises code release but no repository link is provided in the submission; the compute budget is not disclosed.

The evaluation distribution for braid words is partly a contribution of the paper itself, meaning the anti-shortcut construction in Section 4.2 is simultaneously a training protocol choice and the benchmark against which the model is tested. No held-out evaluation under an independently constructed braid distribution is included, and the 14,000 evaluation examples from an infinite non-abelian group represent a negligible fraction of the word space at any given length.


references:

[1] Sung I. Robust Reasoning as a Symmetry-Protected Topological Phase. arXiv preprint arXiv:2601.05240. 2026 Jan 8.

[2] Arjovsky M, Shah A, Bengio Y. Unitary evolution recurrent neural networks. InInternational conference on machine learning 2016 Jun 11 (pp. 1120-1128). PMLR.

[3] Wisdom S, Powers T, Hershey J, Le Roux J, Atlas L. Full-capacity unitary recurrent neural networks. Advances in neural information processing systems. 2016;29.

[4] Nanda N, Chan L, Lieberum T, Smith J, Steinhardt J. Progress measures for grokking via mechanistic interpretability. arXiv preprint arXiv:2301.05217. 2023 Jan 12.

[5] Power A, Burda Y, Edwards H, Babuschkin I, Misra V. Grokking: Generalization beyond overfitting on small algorithmic datasets. arXiv preprint arXiv:2201.02177. 2022 Jan 6.

[6] Lample G, Charton F. Deep learning for symbolic mathematics. arXiv preprint arXiv:1912.01412. 2019 Dec 2.

[7] Charton F. Linear algebra with transformers. arXiv preprint arXiv:2112.01898. 2021 Dec 3.

[8] Delétang G, Ruoss A, Grau-Moya J, Genewein T, Wenliang LK, Catt E, Cundy C, Hutter M, Legg S, Veness J, Ortega PA. Neural networks and the chomsky hierarchy. arXiv preprint arXiv:2207.02098. 2022 Jul 5.

[9] Anil C, Wu Y, Andreassen A, Lewkowycz A, Misra V, Ramasesh V, Slone A, Gur-Ari G, Dyer E, Neyshabur B. Exploring length generalization in large language models. Advances in Neural Information Processing Systems. 2022 Dec 6;35:38546-56.

**Requested Changes:**

Comparative baselines (at minimum a transformer and a multiplicative LSTM trained under identical data and evaluation conditions) must be included with performance reported at all evaluated lengths. All metrics must include variance over at least five independent random seeds with 95 percent confidence intervals. The evaluation sample size per length must be stated explicitly and appropriate confidence intervals for classification metrics must be reported. The framing of "symmetry-protected" properties must either be supported by a formal guarantee or recast as an empirical observation without the protective connotations of topological language. A sensitivity analysis over representation dimension d, margin hyperparameters, and staged training schedules is required to establish robustness of the energy gap. The submission should also be deferred until [1] undergoes peer review or incorporate the necessary derivations, because building the entire inference architecture on an unreviewed preprint makes it impossible to assess the novelty contribution of the present work in isolation.

---

### Review · Reviewer_H48v · 2026-04-16

**Summary Of Contributions:**

This paper solves the algebraic word identity verification problem with exceptionally high accuracy. To address the issue of error accumulation leading to state drift when traditional neural networks process long sequences, the authors first represent each discrete input token as an orthogonal rotation matrix in a high-dimensional space. They then simulate the semantic composition of the symbol sequence through ordered multiplication of these matrices. Finally, they employ a task-agnostic identity-energy decoder that directly measures the distance between the final multiplied matrix and the identity matrix, thereby determining whether the entire sequence of operations is equivalent to "having done nothing at all." Although the method builds upon Holonomic Networks, the authors have adapted and applied it to the algebraic word identity problem with undeniable effectiveness. The main drawback of the paper is the lack of direct comparative experiments against existing methods on this task, both in terms of accuracy and inference efficiency.

**Audience:**

Yes

**Audience Explanation:**

Although this paper focuses on the specific task of algebraic word identity verification, its core methodology—encoding the compositional semantics of symbol sequences as operator composition in a geometric space and performing verification by measuring the distance to the identity element—carries broader implications. Taking LLMs and Agents as examples, both large language models during long-horizon reasoning and agents executing complex tasks encounter the very problem this paper addresses: the tendency to forget initial constraints and suffer from state drift in intermediate steps. In summary, while the paper's experiments are confined to the algebraic domain, the geometric principles it reveals for stabilizing reasoning have clear reference value for LLM reasoning, hierarchical planning, and neuro-symbolic AI more broadly.

**Claims And Evidence:**

Yes

**Claims Explanation:**

The paper reports exceptionally high accuracy in the experimental section, which sufficiently demonstrates the viability of the method. The dataset construction is also presented in considerable detail to verify the correctness of the labels. Furthermore, building upon Holonomic Networks, the authors rigorously articulate their methodological design: discrete tokens are parameterized as orthogonal rotation matrices, ordered matrix multiplication replaces traditional additive state updates, and a task-agnostic identity-energy decoder is introduced to directly measure the distance between the final composed matrix and the identity matrix.

**Requested Changes:**

1. Expand related work for better readability. The current manuscript provides a relatively brief background discussion, which may present a non-trivial barrier to entry for readers less familiar with this specific subfield. Expanding the related work section modestly would help a broader audience establish the necessary context, thereby improving the paper's overall readability and accessibility.

2. Add comparative baselines for accuracy and efficiency. Include comparisons with standard sequence models on the same task. This establishes a difficulty baseline and clarifies the accuracy–efficiency trade-off.

---

### Review · Reviewer_MT6V · 2026-05-05

**Summary Of Contributions:**

The paper introduces a modular Holonomic Network template that achieves perfect algebraic verification on words 100 times longer than training samples for both finite Coxeter and infinite braid group settings.

---

### Strengths:
- The model proposed by the authors maintains accuracy over sequence lengths significantly longer than those encountered during training.
- The paper introduces a "Unified Neural Template" that is modular and reusable across different types of reasoning tasks.

---

### Weaknesses:
- Despite the use of the term 'verified' in the title, the paper does not provide a formal theory or guarantees of the proposed behavior. The claims are supported empirically, but there is no underlying theory explaining why the method works.
- Section 5 lacks competitive baselines. In particular, comparisons against standard architectures (transformer-based models) are necessary to demonstrate that the proposed approach offers a meaningful advantage.
- The method appears to rely on carefully designed anti-collapse penalties (Equation 12) to avoid degenerate identity-matrix solutions; can the authors provide some evidence that indicates that the claimed algebraic reasoning behavior is an intrinsic property of the architecture and NOT only due to a fragile and highly engineered loss design?
- In the braid group experiments, I have a small concern about the use of a small, fixed library (C, Section 4.2) to constrain representations: can the authors verify that the model is not merely overfitting to these specific words, rather than learning generalizable braid algebra?
- Lastly, my biggest criticism is that the paper uses dense technical jargon without sufficient intuition, which significantly harms readability. This makes it difficult for me and potential future readers to follow the core ideas and it weakens the overall impact of the work.

**Audience:**

Yes

**Audience Explanation:**

Readability is a big concern here, but with sufficient changes, the community would definitely be interested in this paper and this work, especially since the authors show very strong empirical evidence of their model working effectively.

**Claims And Evidence:**

No

**Claims Explanation:**

Some baselines are missing, but majority of the claims seem to be supported.

**Requested Changes:**

Most of my concerns are already captured in the listed weaknesses. Regardless, I strongly recommend that the authors revise the paper with clearer, simpler language to improve readability and make the core ideas easier to follow.

---

### Decision · Action_Editor_Ls4r · 2026-06-28

**Recommendation:** Reject

**Additional Comments:**

The two substantive reviews both lean reject (JXKu and MT6V), and their central concerns -- missing baselines, absent statistical evidence, and overstated framing -- are the kind that determine whether the headline claims hold rather than matters of polish. The one leaning-accept review (H48v) is largely a summary of the method; when asked to justify the positive recommendation it was not elaborated, and it too requested the comparative baselines that are missing, so I give it limited weight. The deciding factor is that the authors did not engage with the review process: a revision promised on 18 May never materialized and no reviewer concern was answered, so the gaps identified at submission remain open. The topic is worth pursuing, but in its current form, with the concerns unaddressed, the paper does not meet TMLR's evidentiary bar.

A concrete path to acceptance in a future major revision: (1) add comparative baselines, at minimum a transformer and a multiplicative or unitary recurrent network, trained on identical data and evaluated at all reported lengths; (2) report variance over at least five seeds with 95% confidence intervals, state the per-length evaluation sample size with appropriate confidence intervals for the classification metrics, and add a sensitivity/ROC analysis over the margins, the representation dimension d, and the staged schedule; (3) either support the "symmetry-protected"/"verified" framing with a formal guarantee or recast it as an empirical observation without the protective connotation; and (4) provide a held-out braid evaluation under an independently constructed distribution, release code, and disclose the compute budget and curriculum details needed for reproduction.

Two reviewers (MT6V and H48v) also flagged readability: dense jargon and a thin related-work and background section. A future submission should foreground intuition and motivation alongside the formal development.

**Audience:**

Yes

**Audience Explanation:**

The underlying question, stabilizing compositional computation so that correctness holds under semantics-preserving rewrites and well beyond the training length, is of genuine interest, and encoding compositional semantics as operator composition in a geometric space and verifying by distance to the identity is an appealing idea. Reviewers MT6V and H48v both answered yes, noting potential relevance to long-horizon reasoning and neuro-symbolic methods. Reviewer JXKu answered no, but on the ground that the current evidence does not let the audience draw warranted conclusions, not that the topic itself is uninteresting. The interest is real; the present execution does not yet support it.

**Claims And Evidence:**

No

**Claims Explanation:**

The paper reports perfect verification (TPR = 1, FPR = 0) under a 100x length extrapolation for Coxeter words in S_32 and braid words in B_8, but the evidence offered does not let a reader conclude that these results are caused by the proposed holonomic mechanism, or that they are stable. I weight Reviewer JXKu's assessment most heavily, as it is the most detailed and best-evidenced review, and its central points went unanswered:

1. No comparative baselines. The paper claims standard sequence models suffer "severe drift," but no transformer or recurrent baseline is trained on the same data under the same conditions. Without this, the length-extrapolation result cannot be attributed to the holonomic parameterization rather than to the task being learnable by simpler models. Reviewer JXKu's review further points to a substantial body of relevant prior art -- unitary/norm-preserving recurrent networks, grokking and the mechanistic interpretability of learned group structure, and sequence-model results on symbolic and Chomsky-hierarchy tasks -- that is neither compared against nor, in several cases, cited; I refer to that review for the specific references.

2. No statistical evidence. All headline numbers are single-run point estimates, with no variance across seeds, no confidence intervals, and no statement of how many runs were performed or whether the reported model is best, median, or representative. The energy gap (Delta about 9.4e-2) is presented as a robust certificate, but its variance and its sensitivity to the tunable margins m+ and m- (which also set the threshold tau in Eq. 9) are not examined, and no ROC analysis is given. For the Coxeter setting, the per-length sample size is not stated in the main text, so even a binomial confidence interval for perfect classification cannot be computed by the reader.

3. Overstated framing. The title and "symmetry-protected" language carry a guarantee connotation, but the authors themselves state (Section 2.3) that the framing "summarizes an empirical separation phenomenon" rather than a theorem, and that the guarantees from Sung (2026) hold only "under a realizability condition." Presenting an empirical energy gap as "protection" or "verified" reasoning overstates the theoretical content.

4. Contribution scope and reproducibility. The inference machinery (SO(d) parameterization, holonomic composition, per-step renormalization) originates in an unreviewed arXiv preprint (Sung 2026); the present contribution reduces to substituting a Frobenius-distance-to-identity decoder. No code repository is provided despite the promised release, the staged curriculum is described only qualitatively, the compute budget is undisclosed, and there is no ablation over the representation dimension d or the loss weights in Eq. 12.

5. Benchmark independence. For braids, the anti-shortcut evaluation distribution (Section 4.2) is itself partly a contribution of the paper, so the training protocol and the benchmark are entangled, and no held-out evaluation under an independently constructed braid distribution is included.

Reviewer MT6V independently raised that the algebraic behavior may stem from carefully engineered anti-collapse penalties (Eq. 12) rather than from the architecture, and that the small fixed library C in the braid setting risks overfitting to specific words.

None of these concerns were addressed. The authors posted on 18 May that a revision was forthcoming, but no revision and no point-by-point response were submitted, so the evidentiary gaps stand as raised.

**Resubmission Of Major Revision:**

The authors may consider submitting a major revision at a later time.